# QORA: A Sustainable Framework for Open-World Generative Model Attribution with Quasi-Orthogonal Representation Disentanglement

## Abstract

The rapid emergence of new generative models poses significant challenges to static attribution frameworks, which often confidently misattribute images from unknown sources to known ones and struggle to adapt stably to new models. To address these limitations, we propose Quasi-Orthogonal Representation Attribution (QORA), a unified framework for sustainable open-world generative model attribution. QORA consists of two core modules. The Progressive Orthogonal Learning Module (POLM) employs Stiefel manifold optimization to construct a quasi-orthogonal feature space that reduces redundancy while maintaining a stable attribution subspace for open-world settings. The Fingerprint Disentanglement and Enhancement Module (FDEM) leverages classifier-guided attention and multi-auxiliary contrastive learning to disentangle and amplify model-specific fingerprints. To enable continual learning, QORA integrates exemplar replay with feature-similarity-based classifier initialization, achieving lightweight incremental updates for new models while avoiding catastrophic forgetting. Extensive experiments demonstrate that QORA achieves state-of-the-art closed-set accuracy and strong open-set robustness across GAN and diffusion benchmarks, while maintaining stable performance during incremental learning, highlighting its superior scalability and applicability in evolving environments.

## 1 Introduction

Generative AI has made remarkable progress in image quality, diversity, and controllability, with applications spanning from entertainment to production. Yet these capabilities also raise serious security concerns, as maliciously crafted synthetic images are exploited to spread misinformation, fabricate events, and manipulate public opinion, threatening the integrity of the digital ecosystem. To mitigate risks, leading AI companies have pledged to embed watermarks into generated content (Bartz & Hu), but such active solutions lack universality. This has driven research into passive methods that detect AI-generated content (Wang et al., 2020b; 2023b; Ojha et al., 2023), though they generally fail to identify the specific source model—information critical for responsibility tracing and accountability.

To address this, the task of generative model attribution has been developed to passively trace the source generator. Early reconstruction-based methods (Albright et al., 2019) exploited cross-model reconstruction errors but were limited to GANs. Fingerprint-based approaches later demonstrated distinct model-specific traces (Yu et al., 2019b; Marra et al., 2019a), enabling multi-class attribution (Yang et al., 2021; Bui et al., 2022), while MAID (Zhu et al., 2025) extended attribution to diffusion models. These methods, however, are closed-world and often misattribute images from unseen generators to the nearest known model. Open-world attribution addresses this limitation by combining attribution with rejection of unknown classes, using strategies such as patch-based contrastive learning (Yang et al., 2022), rejection-aware classifiers (Wang et al., 2023a), metric learning (Fang et al., 2023b), feature augmentation (Yang et al., 2023), Siamese verification (Abady et al., 2024), forensic self-descriptions (Nguyen et al., 2025), and frequency-domain masking (Zhang et al., 2025). Despite these advances, most methods are trained on limited data, are sensitive to irrelevant con-

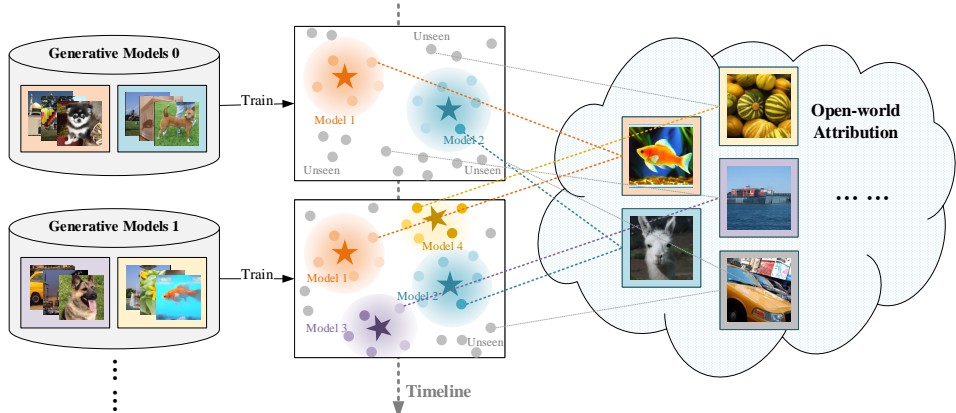

Figure 1: Overview of the SOW-GMA task, which requires the attribution system incrementally incorporates new generative models while maintaining accurate attribution for known sources and reliably rejecting unknown ones, ensuring long-term scalability in dynamic real-world scenarios.

tent or texture, and require full retraining to accommodate new models. As large-scale pretrained Vision–Language Models (VLMs) can produce robust, content-agnostic features, OCC-CLIP (Liu et al., 2024) adapts CLIP for few-shot attribution, while Cioni *et al.* (Cioni et al., 2025) analyze their feature layers for generalization. These approaches, however, typically use embeddings directly and do not optimize VLMs specifically for model attribution task or filter out irrelevant information.

Moreover, the rapid emergence of new generative models further underscores the need for sustainable open-world attribution, as illustrated in Fig. 1. Current solutions suffer from high computational cost, memory overhead, and catastrophic forgetting during incremental updates (Li et al., 2024a). A practical framework must therefore support accurate attribution of known generators, reliable rejection of unknowns, and efficient adaptation to new models without full retraining.

To this end, we propose Quasi-Orthogonal Representation Attribution (QORA), a scalable framework for the Sustainable Open-World Generative Model Attribution (SOW-GMA) task. Built on the CLIP-ViT L/14 backbone, QORA extracts mid-level features containing generative fingerprints and fine-tunes them via LoRA for artifact sensitivity. It introduces the Progressive Orthogonal Learning Module (POLM) to reduce feature redundancy and construct a stable artigact space for the open-world generators, and the Fingerprint Disentanglement and Enhancement Module (FDEM) to isolate and amplify fingerprint-specific signals for closed-set attribution. During incremental learning, QORA freezes most parameters and expands only lightweight classifiers with exemplar replay, enabling efficient adaptation with minimal overhead. The main contributions can be summarized as follows:

- We propose QORA, a practical and scalable framework for SOW-GMA task, which jointly supports accurate closed-set attribution, reliable open-set rejection, and efficient incremental learning for real-world deployment.

- We design a synergistic dual-module architecture, in which POLM construct a stable artigact space for open-world generators, and FDEM decouples and amplifies closed-set model-specific fingerprints.

- We first introduce Stiefel manifold optimization into generative model attribution. By constraining the encoder weights to yield maximally independent feature dimensions that better capture subtle generative fingerprints.

## 2 RELATED WORKS

**Artifacts in AI-Generated Images.** AI-generated images contain visually subtle but detectable artifacts that differ across architectures and can be exploited for attribution. Early studies emphasized frequency-domain traces, such as irregular mid–high-frequency patterns in GAN outputs (Durall et al., 2020), leading to classifiers based on frequency domains (Frank et al., 2020; Jeong

et al., 2022c). However, these methods generalize poorly to diffusion models, whose artifacts are less frequency-pronounced. Recent works shift focus to spatial-domain cues, leveraging shallow-layer textures (Liu et al., 2020; Zhong et al., 2023), residual modeling (Sinitsa & Fried, 2024), or diffusion-specific reconstruction artifacts (Zhong et al., 2025; Wang et al., 2023b). Pretrained VLMs (Ojha et al., 2023; Sha et al., 2023; Zhu et al., 2023) further improve generalization by extracting robust, content-invariant features. Our approach builds on this line by exploiting mid-level VLM features to extract stable spatial-domain fingerprints.

**Generative Model Attribution.** Attribution methods aim to identify the source generator of synthetic images. Active approaches embed watermarks but lack generality, while passive approaches exploit model-specific fingerprints. Recent closed-world methods adopt multi-class classification (Yang et al., 2021; Bui et al., 2022) or reconstruction errors (Albright & McCloskey, 2019; Zhu et al., 2025), but fail to generalize to unseen models. Open-world attribution extends to novel classes through strategies such as transformation-pretrained contrastive learning (Yang et al., 2022), Transformer-based localization (Wang et al., 2023a), metric learning (Fang et al., 2023a), feature-space augmentation (Yang et al., 2023), similarity verification (Abady et al., 2024), and spectral masking (Zhang et al., 2025). Despite these advances, most methods require retraining to handle new models and often struggle to suppress irrelevant content. Our work addresses these limitations by introducing quasi-orthogonal projection to suppress redundancy and construct a stable artifact space, while disentangling fingerprints to achieve sustainable attribution in the open world.

**Category-Incremental Learning for Attribution.** The continual emergence of new generators renders static attribution impractical. Category-Incremental Learning (CIL) (Wang et al., 2024; Ji et al., 2023) expands recognition capacity without full historical data, with prior work exploring contrastive learning (Pan et al., 2023), adapters (Gao et al., 2024), or regeneration-based updates (Li et al., 2024b). For GAN detection, incremental and adapter-based frameworks (Marra et al., 2019b; Tang et al., 2025) alleviate semantic drift. In attribution, however, most solutions remain costly or inflexible. We propose a unified framework that combines open-world rejection with class-incremental expansion, using a compact exemplar memory and feature-similarity-based classifier initialization to achieve scalable, sustainable attribution.

# 3 PROBLEM DEFINITION

The SOW-GMA task is designed for a realistic and dynamic setting where generative models continuously emerge. The objective is to build an attribution framework supporting open-set recognition and sustainable incremental learning. Training proceeds over sessions $t = 0, 1, \ldots, T$, where the model receives a labeled dataset

$$\mathcal{D}_t^L = \{(x_{t,i}, y_{t,i})\}_{i=1}^{N_t}, \quad y_{t,i} \in \mathcal{C}_t^L, \tag{1}$$

where $x_{t,i}$ is a generated image and $y_{t,i}$ its source model label, together with a memory buffer $\mathcal{D}_t^M \subseteq \bigcup_{i=0}^{t-1} \mathcal{D}_i^L$ that stores exemplars from past sessions. The cumulative known classes are $\mathcal{C}_t^K = \bigcup_{i=0}^{t} \mathcal{C}_i^L$.

In addition to labeled data, a continuously growing unlabeled data pool $\mathcal{D}_t^U = \{x_i\}_{i=1}^{m}$ is also available, whose classes $\mathcal{C}_t^U$ may include both known $\mathcal{C}_t^K$ and novel unknown classes $\mathcal{C}_t^N \subseteq \mathcal{C}_t^U \backslash \mathcal{C}_t^K$.

The goal of SOW-GMA is to learn a continually adaptive feature extractor $\phi(\cdot)$ that:

1. attributes generated images from known models to $\mathcal{C}_t^K$,
2. rejects generated images from novel unknown models $\mathcal{C}_t^N$ as out-of-distribution,
3. incorporates new model classes through lightweight updates with limited memory $\mathcal{D}_t^M$.

# 4 METHOD

To address the SOW-GMA task, we propose QORA (Fig. 2), which uses the CLIP-ViT L/14 encoder pretrained on large-scale image–text data to reduce attribution biases. Features are extracted from the 12-th transformer block, and fine-tuned with LoRA for attribution alignment. These features are further processed by POLM and FDEM, POLM enhances open-world attribution generalization, while FDEM strengthens fingerprint discriminability.

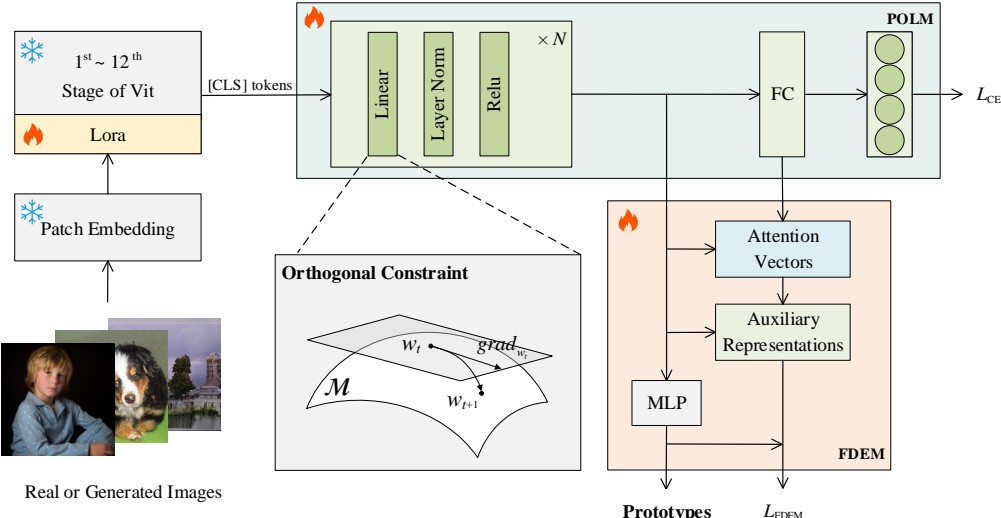

Figure 2: Overview of the proposed QORA framework. CLS tokens are first extracted from a pre-trained CLIP-ViT backbone with LoRA-based fine-tuning. These tokens are transformed by POLM to construct a stable quasi-orthogonal feature space. The FDEM then disentangles and amplifies model-specific fingerprints. After training, class prototypes are obtained by averaging the attribution features produced by FDEM for each category.

## 4.1 POLM

POLM integrates an orthogonally constrained encoder with a dimension-wise normalized classifier to project pretrained features into a quasi-orthogonal subspace. This space reduces redundancy and amplifies subtle artifact cues, providing a stable foundation for fingerprint disentanglement, enhancement, and sustainable incremental attribution.

Specifically, POLM maps the CLS token $\mathbf{f}_{\text{cls}} \in \mathbb{R}^d$ from the ViT encoder into quasi-orthogonal representations via an $N$-layer orthogonally-constrained MLP:

$$\mathbf{f}^{(0)} = \mathbf{f}_{\text{cls}}, \quad \mathbf{f}^{(l)} = \text{ReLU}\left(\text{LN}\left(W_{\text{o}}^{(l)}\mathbf{f}^{(l-1)}\right)\right), \quad \mathbf{f}_{\text{o}} = \mathbf{f}^{(N)}, \tag{2}$$

where $l = 1, 2, \ldots, N$, $W_{\text{o}}^{(l)} \in \mathbb{R}^{d \times d}$ denotes the weight of the $l$-th layer in the encoder, and $\text{LN}(\cdot)$ and $\text{ReLU}(\cdot)$ denote layer normalization and activation. To ensure strict orthogonality, we constrain $W_{\text{o}}$ to lie on the Stiefel manifold (Stiefel, 1935):

$$\mathcal{M}_{d,d} = \left\{W_{\text{o}} \in \mathbb{R}^{d \times d} \mid W_{\text{o}} \cdot W_{\text{o}}^{\top} = I_d\right\} \tag{3}$$

Therefore, this constraint can be reformulated as a Riemannian optimization problem:

$$\min_{W_{\text{o}} \in \mathcal{M}_{d,d}} \mathcal{L}(W_{\text{o}}) = \mathcal{L}_{\text{total}} \tag{4}$$

where $\mathcal{L}_{\text{total}}$ denotes the overall loss function of QORA. Meanwhile, to efficiently update $W_{\text{o}}$ on the manifold, we compute a skew-symmetric matrix $A = \nabla_{W_{\text{o}}}\mathcal{L}W_{\text{o}}^{\top} - W_{\text{o}}(\nabla_{W_{\text{o}}}\mathcal{L})^{\top}$, where $\nabla_{W_{\text{o}}}\mathcal{L}$ is the gradient of the loss. The weight matrix $W_{\text{o}}$ is then updated using the Cayley transform (Li et al., 2020):

$$W_{\text{o}}' = \left(I + \frac{\eta}{2}A\right)^{-1}\left(I - \frac{\eta}{2}A\right)W_{\text{o}} \tag{5}$$

where $\eta$ is the learning rate. This update guarantees that $W_{\text{o}}'$ remains orthogonal and ensures numerical stability throughout training.

In contrast to conventional classifiers that apply class-wise normalization to category vectors, we impose feature-dimension-wise normalization on the classifier weight matrix $W_{fc} \in \mathbb{R}^{C \times d}$:

$$W_{fc}[:, j] \leftarrow \frac{W_{fc}[:, j]}{\|W_{fc}[:, j]\|_2} \quad \forall j \in [1, d] \tag{6}$$

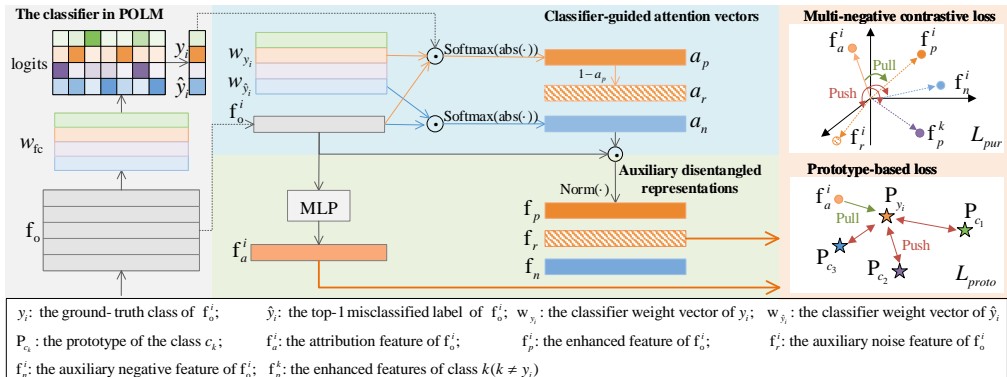

Figure 3: Architecture of FDEM. FDEM disentangles and amplifies generative fingerprints from quasi-orthogonal features produced by POLM. A lightweight MLP projects these features into an attribution space, while classifier weights are used to produce three auxiliary features. Along with class prototypes, these features supervise the attribution learning through contrastive losses.

which balances energy distribution among feature channels, mitigating dominance by high-response channels. This design significantly prevents overfitting to known model categories while enhancing open-set rejection robustness.

## 4.2 FDEM

FDEM enhances the quasi-orthogonal features from POLM by constructing an attribution subspace that leverages class-specific channel importance to isolate and strengthen generative fingerprints. As shown in Fig. 3, given a sample with feature $\mathbf{f}_o^i$ and label $y_i$ (denoted as $\mathbf{f}_o$ and $y$), the attention vectors for the ground-truth class $y$ and the top-1 misclassified class $\hat{y}$ are computed as

$$\mathbf{a}_p = \text{Softmax}\left(\frac{|\mathbf{f}_o \odot \mathbf{w}_y|}{\tau}\right), \quad \mathbf{a}_n = \text{Softmax}\left(\frac{|\mathbf{f}_o \odot \mathbf{w}_{\hat{y}}|}{\tau}\right), \quad (7)$$

where $\odot$ denotes element-wise multiplication and $\tau$ is a temperature parameter.

These attention maps quantify the channel contributions to correct and confusing predictions. Guided by them, we obtain three disentangled representations:

$$\mathbf{f}_{p/r/n} = \text{Normalize}\big(\mathbf{f}_o \odot \mathbf{a}_{p/r/n}\big), \quad \mathbf{a}_r = 1 - \mathbf{a}_p, \quad (8)$$

where $\mathbf{f}_p$ emphasizes discriminative fingerprints, $\mathbf{f}_r$ suppresses irrelevant noise, and $\mathbf{f}_n$ captures misleading fingerprint artifacts.

Thus, FDEM projects $\mathbf{f}_o$ into an attribution space $\mathbf{f}_a$ using a lightweight MLP and optimizes it with a multi-negative contrastive loss:

$$\mathcal{L}_{\text{pur}} = -\log \frac{\exp(\text{sim}(\mathbf{f}_a, \mathbf{f}_p)/\tau)}{\exp(\text{sim}(\mathbf{f}_a, \mathbf{f}_p)/\tau) + \sum_{f \in (\{\mathbf{f}_r, \mathbf{f}_n\} \cup \{\mathbf{f}_p^j\}_{y_j \neq y_i})} \exp\big(\text{sim}(\mathbf{f}_a, f)/\tau\big)}, \quad (9)$$

where $\mathbf{f}_p$, $\mathbf{f}_r$, and $\mathbf{f}_n$ denote enhanced fingerprints, residuals, and confusing artifacts, respectively, $\{\mathbf{f}_p^j\}_{y_j \neq y_i}$ are fingerprints from other classes, $\text{sim}(\cdot, \cdot)$ is cosine similarity, and $\tau$ is a temperature. This formulation aligns $\mathbf{f}_a$ with clean fingerprints while pushing it away from noise, confusions, and unrelated classes.

To further enforce class-level structure, we adopt a prototype-guided loss:

$$\mathcal{L}_{\text{proto}} = -\frac{1}{N}\sum_{i=1}^{N} \log \frac{\exp\big(\mathbf{f}_a^i \cdot \mathbf{p}_{y_i}/\tau\big)}{\sum_{k=1}^{K} \exp\big(\mathbf{f}_a^i \cdot \mathbf{p}_k/\tau\big)} + \frac{1}{K(K-1)} \sum_{j \neq k} \big(\mathbf{p}_j \cdot \mathbf{p}_k\big)^2 \quad (10)$$

where $\mathbf{f}_a^i$ is the normalized attribution feature of sample $i$, $\mathbf{p}_k$ is the prototype of class $k$, $N$ is the number of samples, and $K$ is the number of known classes. The first term enforces intra-class

compactness, and the second prevents prototype overlap. Prototypes are updated by exponential moving average:

$$\mathbf{p}_k \leftarrow (1 - \lambda)\,\mathbf{p}_k + \lambda\,\overline{\mathbf{f}_a^k} \tag{11}$$

where $\overline{\mathbf{f}_a^k}$ is the batch-wise mean attribution feature of class $k$, and $\lambda \in (0, 1)$ is the momentum factor.

Finally, the total training objective with the classifier cross-entropy loss $\mathcal{L}_{\text{CE}}$ from POLM is

$$\mathcal{L}_{\text{total}} = \mathcal{L}_{\text{CE}} + \mathcal{L}_{\text{pur}} + \mathcal{L}_{\text{proto}} \tag{12}$$

During inference, attribution is performed by comparing $\mathbf{f}_a$ to stored prototypes $\mathbf{p}_k$.

### 4.3 SUSTAINABLE INCREMENTAL LEARNING

To integrate new generator classes while preserving performance on previously learned ones, we adopt a memory-efficient incremental learning strategy. In each session $t$, 20 samples per past class are stored in a replay buffer $\mathcal{D}_{t-1}^M$, covering $\mathcal{C}_{0:t-1}^L$. This buffer is then combined with the current session's labeled data $\mathcal{D}_t^L$ of class set $\mathcal{C}_t^L$ to form the updated buffer $\mathcal{D}_t^M$.

During incremental updates, the CLIP-ViT backbone, LoRA parameters, and the POLM encoder are kept frozen. Only the POLM classifier and the MLP in FDEM are updated. Class-wise mean features are first computed for both previously learned classes $k \in \mathcal{C}_{0:t-1}^L$ in $\mathcal{D}_t^M$, and new classes $n \in \mathcal{C}_t^L$ in $\mathcal{D}_t^L$:

$$\overline{\mathbf{f}_o^k} = \frac{1}{|\mathcal{D}_{t,k}^M|} \sum_{\mathbf{f}_{o,i} \in \mathcal{D}_{t,k}^M} \mathbf{f}_{o,i}, \quad \overline{\mathbf{f}_o^n} = \frac{1}{|\mathcal{D}_{t,n}^L|} \sum_{\mathbf{f}_{o,i} \in \mathcal{D}_{t,n}^L} \mathbf{f}_{o,i} \tag{13}$$

where $\mathcal{D}_{t,k}^M$ and $\mathcal{D}_{t,n}^L$ denote the sets of samples belonging to class $k$ and $n$, respectively.

For each new class $n$, the nearest previously known class $k^*$ is identified by

$$k^* = \arg\min_{k \in \mathcal{C}_{0:t-1}^L} \|\overline{\mathbf{f}_o^n} - \overline{\mathbf{f}_o^k}\|_2 \tag{14}$$

and the classifier weight for class $n$ is initialized as

$$\mathbf{w}_n \leftarrow \mathbf{w}_{k^*} \tag{15}$$

Following initialization, incremental training is carried out using the same total loss $\mathcal{L}_{\text{total}}$ as in the initial training phase. After training, updated prototypes are retained for future attribution.

## 5 EXPERIMENT

In this section, we provide a comprehensive evaluation of the proposed QORA framework. We first outline the experimental setups. Then we assess static open-world attribution followed by extensive ablation studies. Finally, we evaluate QORA in a five-session incremental learning scenario, demonstrating its ability of sustainable.

### 5.1 EXPERIMENTAL SETUPS

**Datasets.** We evaluate QORA on two static open-world attribution benchmarks: OSMA (Yang et al., 2023), a GAN-based dataset covering 53 GANs with diverse seeds and architectures, and GenImage (Cioni et al., 2025), a diffusion-based benchmark with ImageNet classes and models from eight diffusion generators. To simulate the continuous emergence of generators, we construct a sustainable open-world benchmark from these datasets. As detailed in in Appendix A, it includes one real-image class and 23 generator classes split into five sessions, each introducing four new seen classes while the unseen set comprises remaining models.

**Evaluation Metrics.** Following established protocols (Yang et al., 2023; Cioni et al., 2025), we evaluate QORA with three metrics: classification accuracy (Acc.) for closed-set attribution of seen

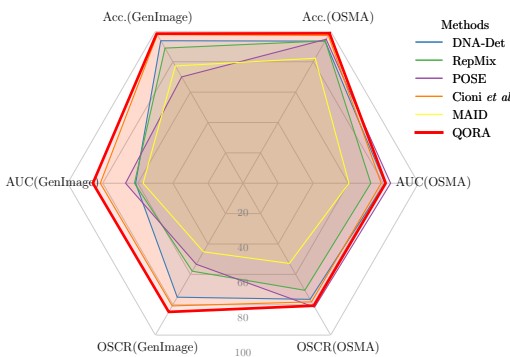

Figure 4: QORA outperforms baselines on OSMA and GenImage in both closed- and open-set metrics, highlighting strong generalization.

Table 1: Performance comparison on the diffusion-based model attribution benchmark GenImage. Results are averaged over five splits. The best performance is shown in bold, and the second-best is underlined.

| Method | Acc. (%) | AUC (%) | OSCR (%) |
|---|---|---|---|
| DNA-Det | 93.83 | 61.27 | 75.08 |
| RepMix | 88.98 | 61.93 | 57.92 |
| POSE | 70.00 | 67.00 | 53.35 |
| Cioni *et al.* | 97.82 | 81.39 | 80.78 |
| MAID | 77.23 | 56.98 | 45.16 |
| QORA | **98.51** | **85.63** | **84.74** |

Table 2: Performance comparison on OSMA. Results are averaged over five splits. The highest score for each metric is shown in bold, and the second-best score is underlined.

| Method | Acc.(%) | Unseen Seed | | Unseen Arch. | | Unseen Data | |
|---|---|---|---|---|---|---|---|
| | | AUC(%) | OSCR(%) | AUC(%) | OSCR(%) | AUC(%) | OSCR(%) |
| PRNU | 55.27 | **69.20** | 49.16 | 70.02 | 49.49 | 67.68 | **48.57** |
| Yu *et al.* | 85.71 | 53.14 | 50.99 | 69.04 | 64.17 | 78.79 | 72.20 |
| DCT-CNN | 86.16 | 55.46 | 52.68 | 72.56 | 67.43 | 72.87 | 67.57 |
| DNA-Det | 93.56 | 61.46 | 59.34 | 80.93 | 76.45 | 66.14 | 63.27 |
| RepMix | 93.69 | 54.70 | 53.26 | 72.86 | 70.49 | 78.69 | 76.02 |
| POSE | 94.81 | 68.15 | **67.25** | **84.17** | **81.62** | 88.24 | 85.64 |
| Cioni *et al.* | 97.29 | 54.15 | 54.00 | 78.78 | 78.12 | 90.60 | 89.52 |
| MAID | 82.30 | 51.06 | 46.02 | 60.40 | 52.81 | **59.04** | 52.01 |
| QORA | **98.68** | 62.56 | 62.23 | 81.34 | 80.66 | 80.68 | 80.08 |

generators, AUC for open-set detection of unseen generators, and OSCR for jointly assessing attribution accuracy and rejection quality in open-world conditions.

**Baseline Methods.** We compare QORA against representative attribution baselines spanning both closed- and open-world settings, including PRNU (Marra et al., 2019a), Yu *et al.* (Yu et al., 2019a), DCT-CNN (Frank et al., 2020), DNA-Det (Yang et al., 2022), RepMix (Bui et al., 2022), POSE (Yang et al., 2023), Cioni *et al.* (Cioni et al., 2025), and MAID (Zhu et al., 2025).

**Implementation Details.** We fine-tune CLIP-ViT L/14 with LoRA with a rank of 16 per adapter, use a one-layer MLP as the POLM encoder, and update FDEM prototypes with a momentum coefficient $\lambda$ of 0.995. Models are trained for 30 epochs on one-quarter of the training data per class using Adam with cosine annealing, where the initial learning rate is set to $1 \times 10^{-4}$. All experiments are implemented in PyTorch 2.0 and run on an NVIDIA RTX 3090.

## 5.2 EVALUATION OF OPEN-SET MODEL ATTRIBUTION

We evaluate QORA against baselines on OSMA and GenImage benchmarks, with overall results summarized in Fig. 4. QORA consistently surpasses prior methods in both closed-set and open-set performance, showing strong generalization across architectures.

**Comparison with SOTA on GAN-generated images.** OSMA evaluates three settings: unseen seeds, unseen architectures, and unseen training data. Strong performance on the first two indicates sensitivity to model-intrinsic fingerprints, while lower performance on unseen data suggests reduced reliance on content semantics. MAID's results were obtained by retraining its open-source implementation under the standard protocol, whereas other baselines are reported from the official OSMA benchmark (Yang et al., 2023). As shown in Table 2, QORA achieves a closed-set attribution accuracy of 98.68%, surpassing the previous best by 1.39%. For open-set evaluation, QORA

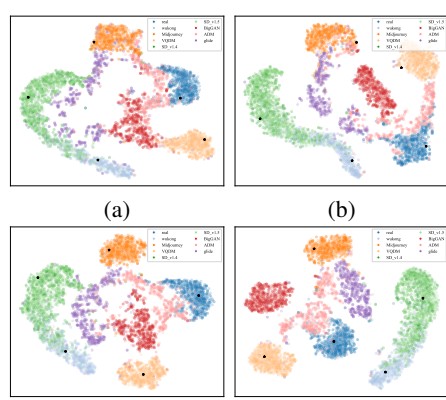

(a)  (b)

(c)  (d)

Figure 5: t-SNE under ablations: (a) w/o or-thogonality or normalization, (b) orthogonality only, (c) normalization only, (d) full setup.

Table 3: Ablation of loss functions over five GenImage splits. The first three rows measure attribution accuracy using attribution represen-tations, while the last row reports classifier per-formance in POLM with pure cross-entropy loss $\mathcal{L}_{\text{CE}}$. The best performance is shown in bold.

| Losses | Acc. (%) | AUC (%) | OSCR (%) |
|---|---|---|---|
| full model | **98.51** | **85.63** | **84.74** |
| w/o $\mathcal{L}_{\text{pur}}$ | 98.27 | 80.33 | 79.22 |
| w/o $\mathcal{L}_{\text{proto}}$ | 96.90 | 82.29 | 80.19 |
| w/o $\mathcal{L}_{\text{pur}}$ and $\mathcal{L}_{\text{proto}}$ | 97.45 | 81.77 | 80.18 |

ranks second on unseen architectures and seeds, trailing POSE, but attains 80.68% AUC and 80.08% OSCR on unseen data, below POSE, which highlights its stronger emphasis on model-intrinsic fin-gerprints rather than semantic variations.

**Comparison with SOTA on diffusion-generated images.** As shown in Table 1, on GenImage, closed-set GAN-specific baselines are excluded due to their limited generalization capability. QORA achieves the highest closed-set accuracy of 98.51%, surpassing Cioni *et al.* by 0.69%. For open-set recognition, it achieves 85.63% AUC and 84.74% OSCR, yielding absolute gains of 4.24% and 3.96% over the previous best. These results demonstrate QORA's effectiveness in capturing discriminative fingerprints of diffusion models.

### 5.3 ABLATION STUDIES

**Ablation Study on POLM.** We assess the contributions of the orthogonality constraint in the en-coder and the dimension-wise normalization in the classifier on split-1 of GenImage. Four config-urations are compared: (a) neither constraint, (b) orthogonality only, (c) normalization only, and (d) both constraints (full QORA). t-SNE visualizations of the attribution features (Fig. 5) show that (a) produces scattered distributions for seen categories and shows clear confusion between seen and unseen features, (b) improves inter-class separation for seen categories, (c) reduces overlap between seen and unseen samples, and (d) achieves well-separated clusters and distinct dispersion of unseen samples, demonstrating enhanced open-set rejection and a stable feature space.

**Ablation on Loss Components in FDEM.** Table 3 evaluates the prototype-guided loss $\mathcal{L}_{\text{proto}}$ and purification contrastive loss $\mathcal{L}_{\text{pur}}$ across all five GenImage splits. Removing either loss degrades performance: without $\mathcal{L}_{\text{proto}}$, closed-set accuracy drops from 98.51% to 96.90%, highlighting its role in aligning features with class prototypes; without $\mathcal{L}_{\text{pur}}$, open-set AUC and OSCR decrease by 5.30% and 5.52%, showing its importance in purifying model-specific fingerprints. When both losses are removed, reliance on the POLM classifier alone leads to further degradation. These results confirm that FDEM is crucial for learning discriminative, generalizable representations and robust open-world attribution under dynamic conditions.

### 5.4 EVALUATION ON SUSTAINABLE OPEN-WORLD ATTRIBUTION

We evaluate QORA on the five-session SOW-GMA benchmark to assess scalability and adaptability under realistic open-world conditions, comparing it with five baselines: DNA-Det (Yang et al., 2022), RepMix (Bui et al., 2022), POSE (Yang et al., 2023), Cioni *et al.* (Cioni et al., 2025), and MAID (Zhu et al., 2025). All models are trained with official implementations, initializing each incremental session from the previous checkpoint.

Table 4: Performance comparison between Session 0 and Session 4 for different methods. The best performance is shown in bold, and the second-best is underlined. Red arrows and text indicate the increase (↑) or decrease (↓) from Session 0 to Session 4.

| Method | Acc. (%) | | AUC (%) | | OSCR (%) | |
|---|---|---|---|---|---|---|
| | Session 0 | Session 4 | Session 0 | Session 4 | Session 0 | Session 4 |
| DNA-Det | 99.16 | 82.39 ↓ 16.77 | 79.33 | 58.78 ↓ 20.55 | 79.80 | 54.92 ↓ 24.88 |
| RepMix | 97.11 | 29.05 ↓ 68.06 | 76.81 | 53.21 ↓ 23.60 | 76.14 | 19.21 ↓ 56.93 |
| POSE | 95.84 | 20.57 ↓ 75.27 | **87.22** | 54.27 ↓ 32.95 | **85.45** | 14.13 ↓ 71.32 |
| Cioni *et al.* | 98.44 | 79.24 ↓ 19.20 | 82.55 | 59.90 ↓ 22.65 | 82.07 | 52.14 ↓ 29.93 |
| MAID | 88.59 | 48.47 ↓ 40.12 | 56.95 | **61.25** ↑ 4.30 | 53.50 | 34.86 ↓ 18.64 |
| QORA | **99.70** | **87.69** ↓ 12.01 | 84.61 | 60.30 ↓ 24.31 | 84.55 | **56.89** ↓ 27.66 |

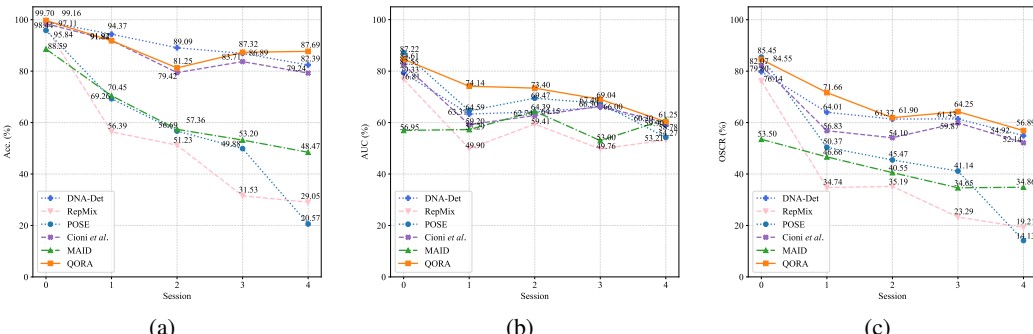

(a)          (b)          (c)

Figure 6: Comparison of four attribution methods over five incremental sessions shows that QORA consistently outperforms others in (a) closed-set accuracy, (b) open-set AUC, and (c) open-set OSCR, demonstrating its superior scalability and stability in open-world incremental learning.

Table 4 reports initial and final session performance. QORA achieves 99.70% closed-set accuracy initially and maintains 87.69% in Session 4, representing the smallest decline with 12.01% among all methods. In contrast, POSE, RepMix, and MAID show sharp degradation of 75.27%, 68.06%, and 40.12%. DNA-Det and Cioni drop to 82.39% and 79.24%, remaining 5–8% below QORA. For open-set detection, QORA's initial AUC and OSCR are slightly lower than POSE's but surpass it by Session 4, with gains of 6.03% in AUC and 42.76% in OSCR. MAID shows large AUC fluctuations, as shown in Fig. 6 (b), whereas QORA consistently keeps AUC above 60%, while other baselines decline or fluctuate. Fig. 6 shows metric trends across sessions. QORA maintains balanced, robust performance in closed- and open-set, effectively integrating new classes while preserving prior knowledge and rejecting unseen generators, demonstrating practical suitability for real-world incremental attribution.

# 6 CONCLUSION

In this paper, we present QORA, a sustainable framework for open-world generative model attribution. Unlike prior methods hindered by emerging models, QORA integrates accurate closed-set attribution, robust open-set rejection, and efficient class-incremental learning with low memory overhead. POLM leverages Stiefel manifold optimization to construct a quasi-orthogonal space that suppresses redundancy and enhances generalization, while FDEM disentangles and strengthens model-specific fingerprints via classifier-guided attention and contrastive learning. A lightweight incremental strategy further supports rapid adaptation without full retraining. Experiments on GAN- and diffusion-based benchmarks show that QORA achieves state-of-the-art attribution accuracy and preserves strong open-set robustness across sessions, highlighting its scalability and real-world applicability.

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

## A  DETAILS OF DATASETS

We evaluate QORA on two **static open-world attribution** benchmarks:

- OSMA Yang et al. (2023): A GAN-based benchmark built on seven real-image datasets, each paired with two GANs for training. Its unseen set includes 53 GANs held out under three conditions: same architecture/dataset with different seeds, novel architectures, and novel training datasets.
- GenImage Cioni et al. (2025): A diffusion-based attribution dataset. Its known classes comprise real ImageNet images and outputs from four diffusion models. Its unseen set consists of samples generated by four additional diffusion models not used during training.

Both benchmarks are evaluated using five train/test splits, with each split varying the composition of seen and unseen generative models to ensure robust generalization testing.

To simulate real-world conditions where generative models continually emerge, we construct a **sustainable open-world attribution** benchmark based on the two datasets described above. As detailed in Table 5, the benchmark includes the real-image class and 20 generative model classes, chronologically divided into five incremental sessions from 2018 to 2022. Session 0 serves as the initial training pshase for the SOW-GMA task. In each session, four newly introduced generative models serve as the session-specific *seen* classes for training. Meanwhile, the *unseen* set comprises all generative models not yet encountered in the current or any previous session, along with three fixed unseen models, SNGAN, S3GAN, and Wav2Lip, that are consistently included in the open-set across all sessions.

As shown in Table 6, the training and testing protocol for each session $t$ is defined as follows:

Table 5: Chronological split of seen and unseen generative models for SOW-GMA task.

| Session | Year | Seen Models | Unseen Models |
|---|---|---|---|
| 0 | 2018 | Real, StarGAN, ProGAN, MMDGAN, BigGAN | SNGAN, S3GAN, Wav2Lip + $\text{Seen}_{1,2,3,4}$ |
| 1 | 2019 | SAGAN, FSGAN, AttGAN, StyleGAN | SNGAN, S3GAN, Wav2Lip + $\text{Seen}_{2,3,4}$ |
| 2 | 2020 | FaceSwap, StyleGAN2, ContraGAN, FaceShifter | SNGAN, S3GAN, Wav2Lip + $\text{Seen}_{3,4}$ |
| 3 | 2021 | StyleGAN3, InfoMaxGAN, ADM, Glide | SNGAN, S3GAN, Wav2Lip + $\text{Seen}_4$ |
| 4 | 2022 | Wukong, Midjourney, Stable Diffusion v1.4, VQDM | SNGAN, S3GAN, Wav2Lip |

Table 6: Data Split for training and testing process.

|  |  | Data Group |
|---|---|---|
| **Train** | **Closed** | $\text{Seen}_t$, $\text{Memory}_t$ |
| **Test** | **Closed** | $\text{Seen}_t$, $\text{Memory}_t$ |
|  | **Open** | $\text{Unseen}_t$ |

- Training: 4K samples are used for each newly introduced class in session $t$, and 20 exemplars are retained for each previously seen class in a memory set denoted as $\text{Memory}_t$.

- Testing: The closed-set includes all classes in $\text{Seen}_t \cup \text{Memory}_t$, while the open-set consists of $\text{Unseen}_t$.

