# OpenReview forum: "QORA: A Sustainable Framework for Open-World Generative Model Attribution with Quasi-Orthogonal Representation Disentanglement"
_ICLR.cc/2026/Conference — ICLR 2026 Conference Withdrawn Submission_

### Official Review · Reviewer_aafq · 2025-10-29

**Soundness:** 2
**Presentation:** 2
**Contribution:** 2
**Rating:** 2
**Confidence:** 3

**Summary:**

The paper proposes QORA (Quasi-Orthogonal Representation Attribution), a framework for attributing generated images to their source models in an open-world setting. The system consists of two main components:  1. POLM (Progressive Orthogonal Learning Module): Creates a quasi-orthogonal feature space using Stiefel manifold optimization. 2. FDEM (Fingerprint Disentanglement and Enhancement Module): Uses classifier-guided attention to isolate and amplify model-specific fingerprints

**Strengths:**

1. This paper addresses a critical need for AI-generated content attribution in an evolving landscape of generative models.

2.  The method supports incremental learning without full retraining, which is crucial for scalability.

**Weaknesses:**

1. The presentation lacks clarity. For example, Critical details are missing in the pipeline illustrated in Figure 1, making it difficult to trace the flow of computation. The input's entry point is ambiguous, and the derivation processes for key components—such as the enhanced feature f _p^i, the auxiliary noise feature f_r^i , and the weight w_y —are not specified.

2. The whole method part only decribes how they are designing objective, but no explanation about why the objectives are nessisary and sufficient.

3. This mixing of residuals, noise, and other class fingerprints appears arbitrary and is not sufficiently motivated. It is uncertain why and how features can be divided into these three types.

**Questions:**

1. How do you actually reject unknown models? What's the threshold?
2. Is there any FLOPs, memory, or inference time comparison?

---

### Official Review · Reviewer_PuiM · 2025-10-30

**Soundness:** 3
**Presentation:** 2
**Contribution:** 3
**Rating:** 2
**Confidence:** 3

**Summary:**

This work proposes Quasi-Orthogonal Representation Attribution (QORA), a unified framework for sustainable, open-world generative-model attribution. QORA comprises two core modules. The Progressive Orthogonal Learning Module (POLM) uses Stiefel-manifold optimization to construct a quasi-orthogonal feature space, reducing redundancy while preserving a stable attribution subspace under open-world shifts. The Fingerprint Disentanglement and Enhancement Module (FDEM) leverages classifier-guided attention and multi-auxiliary contrastive learning to disentangle and amplify model-specific fingerprints. Across GAN and diffusion benchmarks, QORA achieves superiors closed-set accuracy, strong open-set robustness, and stable performance during incremental learning.

**Strengths:**

1. The paper proposes a unified framework for sustainable, open-world generative-model attribution.

2. The introduced Progressive Orthogonal Learning Module (POLM) and Fingerprint Disentanglement and Enhancement Module (FDEM) are well-motivated and technically sound.

3. Extensive experiments cover both GAN and diffusion generators, showing competitive closed-set accuracy and strong open-set performance.

**Weaknesses:**

1. The novelty appears limited for the classifier-guided channel attentions, the orthogonal learning component, and the fingerprint disentanglement, which seem closely related to prior CAM/score-guided attention methods.

2. Report the computational cost of the full framework, especially the Stiefel-constrained MLP, such as FLOPs, parameters, and memory usage.

3. Provide ablation studies for each component of the framework (e.g., POLM, FDEM), including turning individual FDEM losses on/off and sweeping the loss weights used in Eq. 12.

4. Explain the class-prototype update policy in detail.

**Questions:**

1. Please add a comparative analysis situating QORA’s components relative to prior work, for example, clarifying how the classifier-guided channel attentions differ from CAM/score-guided attention.

2. Please include computational cost results and analysis for the full framework (e.g., parameter counts, training/inference time, and memory).

3. Please provide comprehensive ablations covering each component of the framework, including the impact of individual losses and their weights.

---

### Official Review · Reviewer_i1Jj · 2025-11-01

**Soundness:** 3
**Presentation:** 2
**Contribution:** 3
**Rating:** 6
**Confidence:** 2

**Summary:**

The paper propose a novel solution for open-world generative model attribution problem. It is motivated by the generative model attribution problem but with samples from unknown sources as well as continuously coming new models. The new solution features two core modules, Progressive Orthogonal Learning Module (POLM) which reduce the redundancy while make the distance between different source models large and Fingerprint Disentanglement and Enhancement Module (FDEM) to disentangle the model-specific fingerprints.

**Strengths:**

- The paper studies a pressing topic in generative AI era. Generative model attribution (especially under the "open-world" definition) is more realistic than previous settings.
- The experiment shows that the improvement of the new method is obvious.
- The paper also created an experiment setting for continuously coming new models in generative model attribution, which is suitable for following work.

**Weaknesses:**

- The motivation of method design is not clear (or not presented very well). Here are some questions I have so that I may assess the paper in a more accurate way.
  - The high-level reason why we need an orthogonally constrained encoder with a dimension-wise normalized classifier.
  - How does POLM and FDEM intuitively improve the result? Figure 5 and Table 3 are empirical results for ablation study while the gap between different ablation settings are small.
  - Figure 3 is very hard to read or understand the design motivation. One suggestion is to include a small subsection in Section 3 right after the problem definition to introduce how the problem is solved in previous work (with some formula), and describe the weakness as well as the research gap. After all these, introducing the proposed method could be mush easier to read.

**Questions:**

See weakness.

---

### Official Review · Reviewer_SmBx · 2025-11-01

**Soundness:** 2
**Presentation:** 2
**Contribution:** 2
**Rating:** 4
**Confidence:** 5

**Summary:**

This paper introduces QORA, a practical and sustainable framework for open-world generative model attribution. QORA combines a Progressive Orthogonal Learning Module (POLM), leveraging Stiefel manifold optimization to enforce quasi-orthogonal embeddings, and a Fingerprint Disentanglement and Enhancement Module (FDEM) that disentangles and amplifies model-specific generative fingerprints using classifier-guided attention and contrastive learning. QORA supports efficient incremental learning via exemplar replay and classifier initialization, without retraining the backbone. Extensive experiments on GAN- and diffusion-based benchmarks verify the efficacy of the proposed method.

**Strengths:**

1. The dual-module architecture is thoughtfully designed: POLM enforces quasi-orthogonality with Stiefel manifold optimization to reduce redundancy and stabilize the attribution subspace. FDEM proposes a concrete disentanglement mechanism using classifier-guided attention maps and a contrastive, prototype-centric loss.

2. The integration of lightweight, memory-efficient incremental updates via exemplar replay and feature-similarity-based classifier initialization allows QORA to scale with emerging models while mitigating catastrophic forgetting.

3. The extensive experiments validate the effectiveness of the proposed modules and demonstrate their efficiency in handling incremental learning tasks.

**Weaknesses:**

1. The motivation behind the module design in the paper is unclear. For instance, the design of the quasi-orthogonal space lacks a strong connection to the source attribution task. The authors claim that this design reduces redundancy, but no corresponding experiments or visualizations are provided to validate this assertion.

2. The experimental details are inadequately described. In the comparison presented in Table 4, how were the baseline methods fine-tuned for incremental learning scenarios? Were they simply retrained by directly combining new data with the original dataset?

3. The ablation study is insufficient, as it lacks experiments on key hyperparameters such as the coefficient of the exponential moving average. Moreover, the FDEM module proposes two types of negative sample representations, how does each contribute to the performance of the method?

**Questions:**

1. Can you elaborate on the degree to which QORA’s gains stem from the CLIP backbone versus the custom POLM or FDEM modules? Have you considered alternative backbones or initializations for true ablation? Please present additional evidence if available.

2. Regarding the claim about the quasi-orthogonal space, could you provide additional validation beyond final performance metrics, such as dedicated ablation experiments or visualization analyses, to substantiate the alleged reduction in redundancy?

3. For the Stiefel manifold optimization and Cayley transform, could you describe computational cost, convergence rate, and robustness to hyperparameters or initialization? Is there a principled justification for the chosen learning rates and orthogonality penalties?

---

### Note · Authors · 2025-12-17

I have read and agree with the venue's withdrawal policy on behalf of myself and my co-authors.